# Continued attendance in a PrEP program despite low adherence and non-protective drug levels among adolescent girls and young women in Kenya: Results from a prospective cohort study

**Jean de Dieu Tapsoba**[1☯‡], **Jane Cover**[2☯‡], **Christopher Obong'o**[3☯‡], **Martha Brady**[2], **Tim R. Cressey**[4], **Kira Mori**[1], **Gordon Okomo**[5], **Edward Kariithi**[6], **Rael Obanda**[3], **Daniel Oluoch-Madiang**[3], **Ying Qing Chen**[1], **Paul Drain**[7], **Ann Duerr**[1]*

1 Vaccine and Infectious Disease Division, Fred Hutchinson Cancer Research Center, Seattle, Washington, United States of America, 2 PATH, Seattle, Washington, United States of America, 3 PATH, Kisumu, Kenya, 4 Chiang Mai University, Chiang Mai, Thailand, 5 Ministry of Health, Homa Bay, Kenya, 6 PATH, Nairobi, Kenya, 7 Department of Global Health, University of Washington, Seattle, Washington, United States of America

☯ These authors contributed equally to this work.
‡ These authors share first authorship on this work.
* aduerr@fredhutch.org

**Data Availability Statement:** The dataset used for this analysis have been uploaded into Harvard

## Abstract

### Background

In sub-Saharan Africa (SSA), adolescent girls and young women (AGYW) ages 15 to 24 years represent <10% of the population yet account for 1 in 5 new HIV infections. Although oral pre-exposure prophylaxis (PrEP) with tenofovir disoproxil fumarate/emtricitabine (TDF/FTC) can be highly effective, low persistence in PrEP programs and poor adherence have limited its ability to reduce HIV incidence among women.

### Methods and findings

A total of 336 AGYW participating in the PEPFAR-funded DREAMS PrEP program in western Kenya were enrolled into a study of PrEP use conducted between 6/2019 to 1/2020. AGYW, who used daily oral TDF/FTC, completed interviews and provided dried blood spots (DBS) for measurement of tenofovir-diphosphate (TFV-DP) concentrations at enrollment and 3 months later, and 176/302 (58.3%, 95% confidence interval [95% CI 52.3 to 63.8]) met our definition of PrEP persistence: having expressed intention to use PrEP and attended both the second interview and an interim refill visit. Among AGYW with DBS taken at the second interview, only 9/197 (4.6%, [95% CI 1.6 to 7.5]) had protective TFV-DP levels (≥700 fmol/punch) and 163/197 (82.7%, [95% CI 77.5 to 88]) had levels consistent with no recent PrEP use (<10 fmol/punch). Perception of being at moderate-to-high risk for HIV if not taking PrEP was associated with persistence (adjusted odds ratio, 10.17 [95% CI 5.14 to 20.13], $p < 0.001$) in a model accounting for county of residence and variables that had

Dataverse (https://doi.org/10.7910/DVN/IQA4OJ) . It contains the data collected as part of this study (Interviews and TFV-DP levels) as well as selected variables from the DREAMS dataset pertaining to PrEP initiation and refill visits.

**Funding:** This study was funded by the Eunice Kennedy Shriver National Institute of Child Health and Human Development (https://www.nichd.nih.gov) grant numbers R01HD094682 (YC and AD) and 3R01HD094682-02S1 (YC). The funders had no role in study design, data collection and analysis, decision to publish, or preparation of the manuscript.

**Competing interests:** The authors have declared that no competing interests exist.

**Abbreviations:** AGYW, adolescent girls and young women; AUDIT, alcohol use disorder identification test; CI, confidence interval; DBS, dried blood spot; IPV, interpersonal violence; IQR, interquartile range; MSM, men who have sex with men; ODK, Open Data Kit; PrEP, pre-exposure prophylaxis; SD, standard deviation; SSA, sub-Saharan Africa; TDF/FTC, tenofovir disoproxil fumarate/emtricitabine; TFV-DP, tenofovir-diphosphate.

$p$-value <0.1 in unadjusted analysis (age, being in school, initiated PrEP 2 to 3 months before the first interview, still active in DREAMS, having children, having multiple sex partners, partner aware of PrEP use, partner very supportive of PrEP use, partner has other partners, AGYW believes that a partner puts her at risk, male condom use, injectable contraceptive use, and implant contraceptive use). Among AGYW who reported continuing PrEP, >90% indicated they were using PrEP to prevent HIV, although almost all had non-protective TFV-DP levels. Limitations included short study duration and inclusion of only DREAMS participants.

## Conclusions

Many AGYW persisted in the PrEP program without taking PrEP frequently enough to receive benefit. Notably, AGYW who persisted had a higher self-perceived risk of HIV infection. These AGYW may be optimal candidates for long-acting PrEP.

## Author summary

### Why was this study done?

- Women in sub-Saharan African bear a disproportionate burden of new HIV infection; adolescent girls and young women (AGYW) account for 1 in 5 new infections.

- Oral pre-exposure prophylaxis (PrEP) is highly effective for HIV prevention in men who have sex with men, but has been less successful among young women, in large part due to low adherence.

- A better understanding of barriers and facilitators to oral PrEP uptake and adherence is critical to designing effective interventions for AGYW.

### What did the researchers do and find?

- We studied persistence and adherence to oral PrEP among 336 AGYW ages 18 to 24 enrolled in the PEPFAR-funded DREAMS PrEP program and residing in Kisumu and Homa Bay Counties, Kenya.

- We interviewed girls at 2 time points, capturing self-reported information about PrEP use. We collected dried blood spots (DBS) for measurement of a PrEP metabolite (TFV-DP).

- A total of 176 AGYW persisted in the PrEP program, that is, they attended the follow-up and PrEP refill visits and stated that they were continuing PrEP. However, most were not adherent. Only 28/176 (16%) had detectable TFV-DP levels at Interview 2, with only 7 (4%) having levels consistent with protection (≥700 fmol/punch).

- Compared to AGYW who discontinued PrEP altogether, this persistent but non-adherent group ($N = 144$) was older (twice as likely to be ≥22 years), 4 times more likely to be active in the DREAMS program, 10 times more likely to think they would be at

moderate-to-high risk for HIV if not taking PrEP, and 3 times more likely to use injectable contraceptives.

### What do these findings mean?

- While many AGYW intend to use PrEP and come to program visits, they do not take sufficient oral PrEP to provide protection against HIV infection (intention–action gap).

- A more nuanced understanding of this intention–action gap is needed to develop effective programmatic responses.

- There is a need to continue to adjust HIV prevention programs to include new tools, such as long-acting PrEP, to more appropriately meet the needs of AGYW, a population at high-risk for HIV.

## Introduction

While remarkable progress has been made in reducing overall HIV incidence, young women still bear a disproportionate burden of new infections [1,2]. In sub-Saharan Africa (SSA), adolescent girls and young women (AGYW) represent less than 10% of the population yet account for 1 in 5 new HIV infections [1]. Oral pre-exposure prophylaxis (PrEP) with tenofovir disoproxil fumarate/emtricitabine (TDF/FTC) has proved to be a highly effective HIV prevention intervention. Clinical trials and program settings have demonstrated roughly 95% efficacy in men who have sex with men (MSM) who adhere to daily or on-demand dosing [3–7]. In contrast, daily oral PrEP for HIV prevention among women has been less successful, in large part due to low adherence [8–10]. Obstacles including lack of partner support, need to conceal PrEP use from partners, lack of privacy from family members, and low empowerment hinder effective daily oral PrEP use in women, particularly in SSA [11,12].

Injectable long-acting cabotegravir (CAB-LA) provided high-level protection in clinical trials in both MSM and transgender women in the Americas, Asia, and Africa [13] and women in SSA [14] and was found to offer superior HIV prevention relative to oral daily TDF/FTC. In the study in SSA, analysis of dried blood spot (DBS) samples from women in the oral TDF/FTC arm, throughout 1.2 years median follow-up in this trial, showed undetectable levels in 38%, and only 18% with concentrations consistent with 4+ doses/week. That is, women were persistent in visit attendance but not adherent to daily pill use. In contrast, injection coverage was 93% of person-years.

These results expand the possibilities for more effective PrEP use among women. Importantly, many women in clinical trials of daily oral PrEP in SSA continued to attend study visits but did not adhere to daily dosing. Thus, this new modality provides an avenue for more successful HIV prevention, much as implants and injectable contraception offer greater success for pregnancy prevention among women in SSA [15]. A critical next step will be bridging the CAB-LA clinical trial results to real-world settings, especially since prior studies suggest that PrEP visit continuation among AGYW is much higher in clinical trials than in programmatic settings (public and private clinics) [16].

Understanding women who are persistent but not adherent and providing them with long-acting formulations may be a more effective way to reduce population-level HIV incidence.

Data from programmatic settings are important for developing effective implementation strategies for these women but such data very rarely include both program persistence, which focuses on ongoing visit attendance [17–19] and objectively measured adherence. To address this, we leveraged data from the PEPFAR-funded DREAMS initiative which offers daily oral PrEP to AGYW within a program that supports HIV prevention activities. Nesting a study in this context allowed collection of data on PrEP persistence as well as self-reported and objectively measured adherence in a program setting.

Our analysis focuses on Kisumu and Homa Bay Counties of western Kenya, where PATH implements the DREAMS Initiative. About 10,500 new HIV infections were reported in young people ages 15 to 24 years in these 2 counties in 2015, accounting for 14% of the national HIV incidence [20]. As part of the DREAMS implementation, an extensive monitoring and evaluation system (DREAMS database) collects comprehensive data on each AGYW and her participation in the DREAMS program, including data on PrEP initiation and refill visits.

By November 2017, an estimated 1,060 AGYW in Kisumu and Homa Bay Counties had enrolled in the DREAMS PrEP program. The current analysis examines the PrEP use cascade as well as predictors of PrEP persistence and adherence (by self-report and drug levels) in a subset of these AGYW enrolled in a prospective study. The main questions examined were the extent of program persistence (a measure focusing on visit attendance) and the level of protection from HIV infection among program attendees as assessed by measuring PrEP metabolites in blood.

## Methods

### Study design, participant eligibility, and sample size

A prospective study was conducted from 6/2019 to 1/2020 among AGYW participants in the DREAMS PrEP program, which had the following eligibility criteria (1 or more): (1) a sero-discordant relationship (HIV+ partner); (2) a sexual partner of unknown HIV status; (3) transactional sex; (4) a recently diagnosed STI; (5) recurrent use of PEP; (6) sexual activity while taking alcohol or drugs; or (7) injection drug use. This study included face-to-face interviews at enrollment and 3 months later, and DBS sample collection for assays of intracellular drug metabolites (tenofovir diphosphate, TFV-DP, as an objective measure of PrEP adherence) from AGYW who reported PrEP use at the time of interview. Interviews captured self-reported adherence and factors associated with adherence and persistence among AGYW. These data were combined with PrEP program data, including refill data from the DREAMS database.

From the DREAMS database, we identified 613 AGYW who met our eligibility criteria of self-reported age between 18 to 24 years, residing in Kisumu and Homa Bay Counties, enrolled in the PrEP program for 2 to 9 months with a visit for PrEP initiation or refill between October 2018 and April 2019, and had returned for a refill in the 2 months prior to study start in June 2019 (to exclude AGYW who might have recently stopped using PrEP).

From the 613 eligible AGYW, we randomly sampled (using the "sample" command in STATA) to select and enroll 359 AGYW who met the above eligibility criteria; were able to speak English, Dholuo, or Kiswahili; and had indicated on the DREAMS enrollment consent form willingness to be contacted for future studies. This sample size assumed a discontinuation or loss to follow-up rate of 50% and a 2-level factor potentially associated with PrEP adherence. Our sample size was sufficient to detect a minimum absolute difference of 15% in PrEP adherence rates against a binary co-factor (e.g., in PrEP support group versus not in PrEP support group) at 80% power given a type I error rate of 5% and a 2-sided test.

This analysis focuses on 2 study objectives: measurement of PrEP adherence by self-report and biomarkers, and identification of factors associated with PrEP persistence and adherence. Because of very low levels of adherence, we were unable to identify associations with that outcome, as anticipated in the prospective protocol (S1 Protocol) of the study, and concentrated primarily on factors associated with persistence. The final analysis included adjustment for clustering at the ward level (a small administrative unit), which was not included in the original protocol.

Participants provided written informed consent to participate. The study was approved by the Institutional Review Board (IRB) at the Kenya Medical Research Institute (KEMRI), Fred Hutchinson Cancer Research Center, and PATH Research Ethics Committee.

## Study procedures and variables

Research assistants received training in informed consent and research ethics, study procedures, interviewing techniques, and DBS sample collection and storage. We conducted one-on-one in-person structured interviews with all participants in their preferred language, entering data electronically via Open Data Kit (ODK). The instrument captured information on: (1) sociodemographic characteristics; (2) participation in and perceptions of the DREAMS program; (3) experience with PrEP and support for PrEP use among partners, family members, and the community; (4) self-reported continued use of and adherence to PrEP; (5) HIV risk perception; (6) condom and contraceptive use; (7) alcohol use (alcohol use disorder identification test (AUDIT) [21]); (8) intimate partner violence (HITS, 4 items [22,23]); (9) social support (MOS Social Support Scale, 19 items [24–27]); (10) depression (PHQ-9, 10 items [28–30]) and used scales previously employed in Kenya or other African countries [21,23,25–27,29].

A research assistant collected a DBS sample for measurement of intracellular TFV-DP levels from participants who reported they were currently taking PrEP. DBS were collected on Whatman Protein Saver 903 cards and stored in a CDC lab in Kisumu County at −70˚C until shipment for testing at Chiang Mai University, Thailand.

At the follow-up visit (Interview 2) 3 to 4 months post-enrollment (Interview 1), we collected a second DBS sample from those who reported they were continuing PrEP. In all but a few cases, the research assistant who conducted the first interview also conducted the follow-up interview. TFV-DP was analyzed using validated liquid chromatography mass spectrometry (LC-MS/MS) with calibration curve range of 200 to 10,000 fmol/3mm punch. Internal/external quality control samples were included in each run, with external samples cross-validated with intra-laboratory testing. In addition to the study data collected at the 2 interviews, pharmacy PrEP refill information was obtained from the DREAMS program data.

The main outcome of interest in this study is PrEP persistence. Persistent AGYW were defined as having attended both interviews as well as at least 1 interim PrEP refill visit and stating that they were taking PrEP at both interviews. A secondary outcome is PrEP adherence based on the TFV-DP levels from the testing of the DBS samples. Adherent AGYW were defined as having TFV-DP levels of 700+ fmol/punch, the equivalent of 4+ doses per week taken regularly [31]. We included cutoffs at 350 fmol/punch, (approximately 1 to 2 days per week) [31], as well as the lower limit of quantification (200 fmol/punch) and the lower limit of detection (10 fmol/punch); levels below 10 were taken as consistent with no PrEP use in the recent past [32].

## Statistical analysis

We summarized categorical variables as frequency and percentage, and continuous variables as mean and standard deviation (SD) or median and interquartile range (IQR). We used

univariable and multivariable (adjusted) generalized estimating equations modification of logistic regression models accounting for clustering of AGYW within wards to assess the associations between persistence and factors measured at Interview 2. The multivariable models adjusted for county of residence and factors that had a $p$-value <0.1 in the univariable analysis. AGYW who were lost to follow-up or censored due to study closure prior to Interview 2 were excluded from the analysis. We focused the analysis on factors associated with persistence at Interview 2; some factors potentially influencing PrEP adherence and PrEP persistence were measured at Interview 2 only. The proportion of AGYW who attended Interview 2 and had missing information on factors of interest was <10%. The analyses were based on complete case data assuming missing completely at random for the missing data mechanism. A $p$-value ≤0.05 was considered statistically significant for a 2-sided test. All analyses were performed using SAS Version 9.4 (SAS Institute, Cary, North Carolina, United States of America) and graphs were plotted using R version 4.1.0. This study is reported per the Strengthening the Reporting of Observational Studies in Epidemiology (STROBE) guideline (S1 STROBE Checklist).

### Role of funding source

The funders were not involved in the study design or execution, data collection, data analysis, data interpretation, or writing of the manuscript.

### Ethics committee approval

Ethical approvals for the study were obtained from the Institutional Review Board (IRB) at the Kenya Medical Research Institute, PATH, and Fred Hutchinson Cancer Research Center. Written informed consents were obtained for all the DREAMS participants enrolled in the study.

## Results

This study recruited 359 AGYW who were enrolled in the DREAMS PrEP program. Of the 359 AGYW, 336 reported currently taking PrEP and provided a DBS at Interview 1; these 336 are the focus of this analysis. Their mean age recorded in DREAMS program data was 22 years; roughly half (45%) resided in Kisumu County, the remainder in Homa Bay County. At Interview 1, AGYW had been in the DREAMS program for 2.5 years on average and in the PrEP program for a median of 112 (IQR: 90,172) days; the majority (176/336, 52.4%) had initiated PrEP use 4 to 6 months earlier, and 42% were living with their partner, 16% reported multiple sexual partners, 37% believed their partner had other partners, 49% believed their partner's behavior put them at risk, 73% reported inconsistent or no condom use, and 29% reported that their partner's status was HIV+ or unknown (Table 1).

**Table 1. Characteristics of 336 study participants at the first interview.**

| Characteristic | N | Col % |
|---|---|---|
| County of residence, Kisumu | 152 | 45.2 |
| County of residence, Homa Bay | 184 | 54.8 |
| Education | | |
| Primary | 110 | 32.7 |
| Secondary | 186 | 55.4 |
| Postsecondary | 40 | 11.9 |
| Currently in school | 122 | 36.3 |
| In PrEP support group | 263 | 78.3 |
| Currently have a sexual partner | 334 | 99.4 |

(*Continued*)

**Table 1.** (Continued)

| Characteristic | N | Col % |
|---|---|---|
| 1 partner | 281 | 83.6 |
| >1 partner | 53 | 15.8 |
| Married/cohabiting | 144 | 42.9 |
| One or more children | 235 | 69.9 |
| Live with parents or grandparents | 162 | 48.2 |
| Live with partner | 141 | 42.0 |
| Partner has other partners | 125 | 37.2 |
| AGYW believes partner puts her at risk | 163 | 48.5 |
| Contraceptive use | 277 | 82.4 |
| Oral contraceptive | 13 | 3.9 |
| Injectable | 60 | 17.9 |
| Implant | 100 | 29.8 |
| Male condoms | 97 | 28.9 |
| Female condoms | 3 | 0.9 |
| Other | 4 | 1.2 |
| Condom, consistent use | 90 | 26.8 |
| Condom, inconsistent/no use | 246 | 73.2 |
| Partner is aware of PrEP use | 198 | 58.9 |
| Partner is very supportive of PrEP use | 131 | 39.0 |
| Partner is very supportive of PrEP use among those with partner being aware | 131 | 66.2 |
| Partner has an unknown HIV status | 83 | 24.7 |
| Partner is HIV positive | 14 | 4.2 |
| Partner's HIV status is positive/unknown | 97 | 28.9 |
| Experience of intimate partner violence (HITS IPV scale, IPV score >10)* | 56 | 16.7 |
| Harmful or hazardous drinking (AUDIT score ≥8)** | 10 | 3 |
| Depression (PHQ-9) | | |
| No depressive symptoms | 116 | 34.5 |
| Mild depression | 142 | 42.3 |
| Moderate or major depression | 78 | 23.2 |
| Social support (most or all the time, overall social support score > 75%)*** | 40 | 11.9 |
| Months since PrEP initiation at interview 1, 2–3 months | 87 | 25.9 |
| Months since PrEP initiation at Interview 1, 4–6 months | 176 | 52.4 |
| Months since PrEP initiation at Interview 1, 6+ months | 67 | 19.9 |

*Scores of 10 or greater on the 4-question HITS scale ("How often does your partner: physically hurt you, insult you or talk down to you, threaten you with harm, and scream or curse at you?") are consistent with domestic violence [22].

**AUDIT was used to measure potential alcohol abuse. A score of 8 or more is associated with harmful or hazardous drinking, a score of 13 or more in women, and 15 or more in men, is likely to indicate alcohol dependence.

***Social support was measured with the MOS Social Support Survey [24] measuring receipt of support using a Likert scale where 1, 2, 3, 4, and 5 are none, a little, some, most, and all the time, respectively. Items in the 4 subscales (emotional/informational support, tangible support, affectionate support, positive social interaction) were combined by calculating the average of (1) the scores for all 18 items included in the 4 subscales, and (2) the score for the one additional item. Scale scores were transformed to a 0–100 scale using the following formula

$$100 \times \frac{(observed\ score - minimum\ possible\ score)}{(maximum\ possible\ score - minimum\ possible\ score)}.$$

A score of 75% or greater corresponds to having overall social support (averaging across all questions) most or all the time (average score of 4 or higher).

AGYW, adolescent girls and young women; AUDIT: alcohol use disorder identification test; IPV, interpersonal violence; MOS, Medical Outcomes Study; PrEP, pre-exposure prophylaxis.

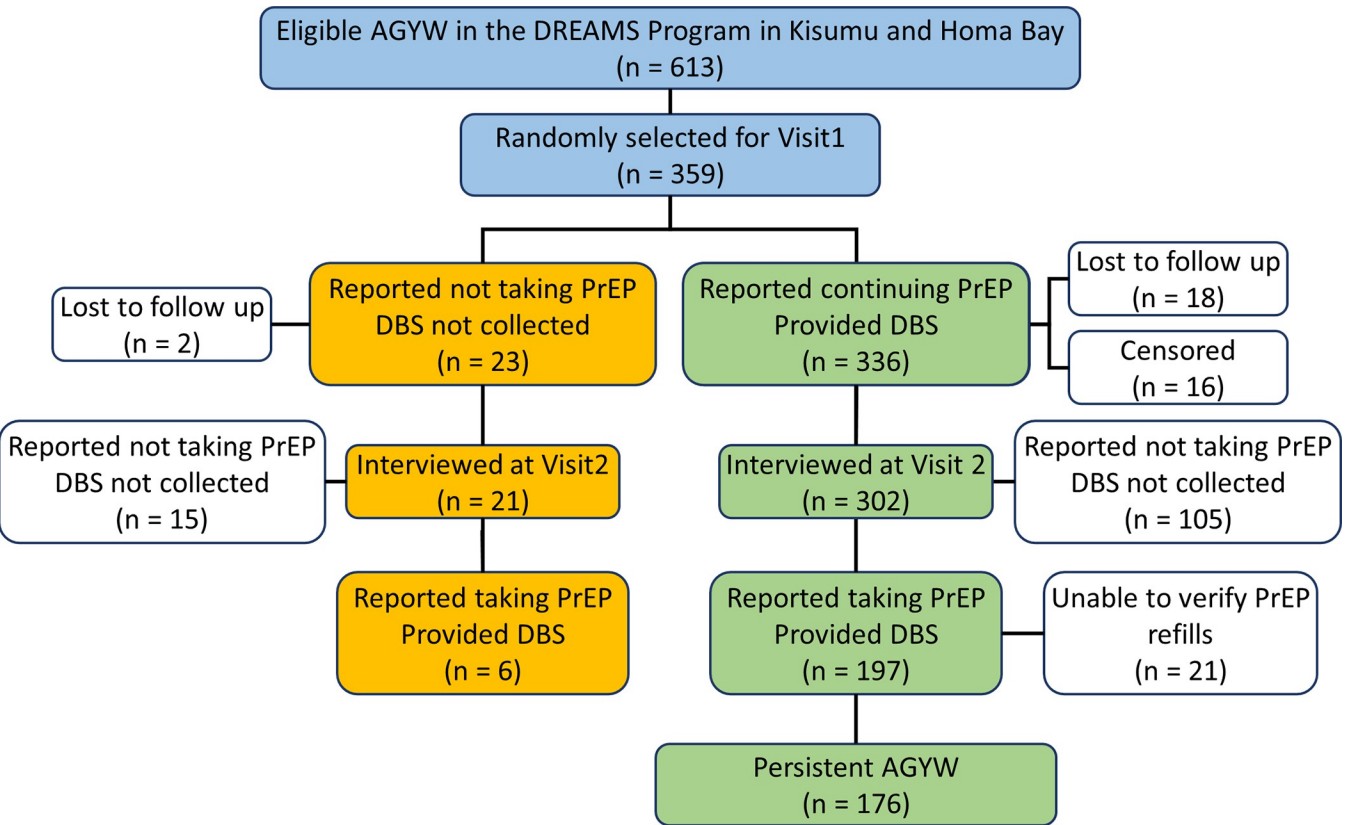

**Fig 1. Study flow chart showing study design and the attrition of participants at each step of the analysis.** Most (246/302, 81%) AGYW interviewed at follow-up had attended at least 1 PrEP dispensation visit between the 2 interviews. Fifty-six had no record of attendance at interim PrEP refill visits, including 21 of the 197 AGYW who reported they were continuing PrEP. This left 176 of the AGYW who attended Interview 2 who met our definition of PrEP persistence (176/302, 58.3%, [95% CI 52.7–63.8] (Figs 1 and 2A and Table A in S1 Table). AGYW, adolescent girls and young women; CI, confidence interval; DBS, dried blood spot; PrEP, pre-exposure prophylaxis.

Of the 336 AGYW taking PrEP at enrollment, 302 attended Interview 2, with 18 lost to follow-up and 16 censored. At Interview 2, 105 said they had discontinued PrEP and 197 said they were continuing PrEP (Fig 1).

The median time between the 2 interviews was 93 (IQR [91, 99]) days. Among AGYW whose TFV-DP levels at enrollment indicated they were taking at least some PrEP (>10 fmol/punch), a high proportion were persistent (52/78: 66.7%, [95% CI 56.2 to 77.1]) (Fig 2B, Table B in S1 Table).

While visit attendance suggested high persistence (Figs 1 and 2B) and self-reported PrEP use was high, actual adherence was low with very few AGYW taking sufficient PrEP to be protected from HIV acquisition. At Interview 1, 298/336 (89%, [95% CI 85 to 92.1]) AGYW reported taking 4 or more (4+) doses in the past 7 days. However, only 21/336 (6.3%, [95% CI 3.7 to 8.8]) had TFV-DP levels consistent with 4+ doses per week (700+ fmol/punch). In fact, although only 20 AGYW (6%, [95% CI 3.4 to 8.5]) reported taking 1 or no doses in the past week, 258/336 (76.8%, [95% CI 72.3 to 81.3]) had TFV-DP levels <10 fmol/punch, which is consistent with no doses in the recent past. Results at Interview 2 were similar. Although most (166/197, 84%, [95% CI 79.2 to 89.3]) of the AGYW who had samples taken at this visit reported taking 4+ doses in the past 7 days, only 9 (4.6%, [95% CI 1.6 to 7.5]) had TFV-DP levels consistent with this level of adherence and 163 (82.7%, [95% CI 77.4 to 88]) had levels consistent with no PrEP use.

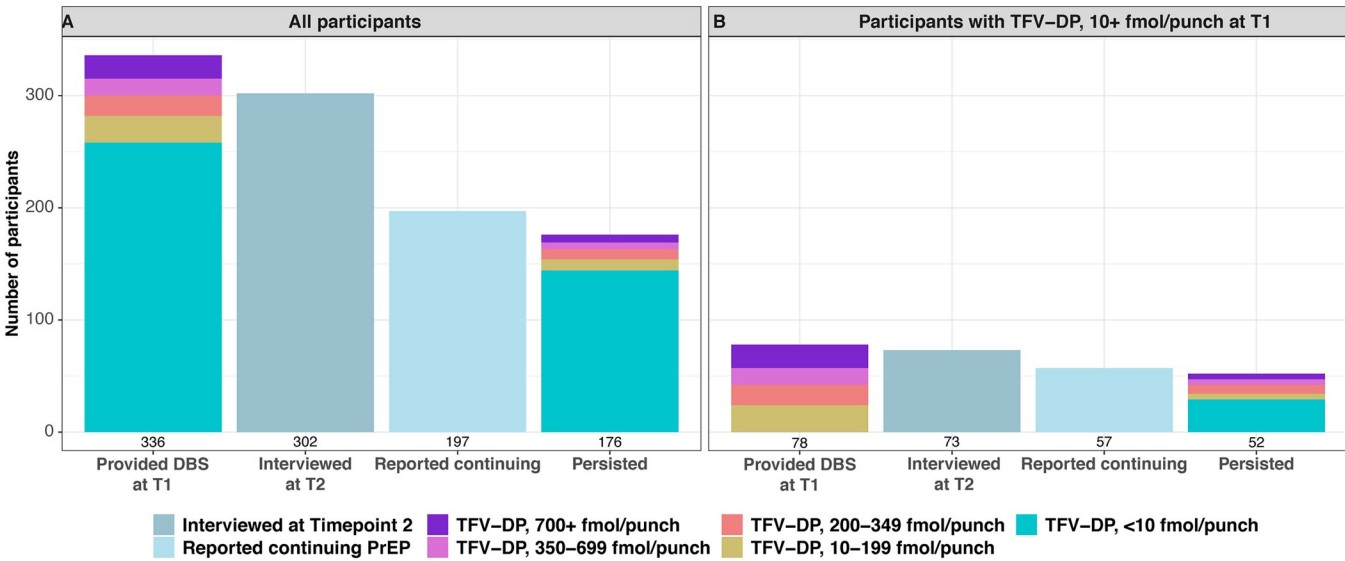

**Fig 2. PrEP Cascade.** (A) All participants. (B) Participants with detectable TFV-DP levels (10+ fmol/punch) at the first Interview. T1: first interview visit (Time point 1). T2: second interview visit (Time point 2). The numbers of participants in the categories of TFV-DP levels are shown in S1 Table. PrEP, pre-exposure prophylaxis; TFV-DP, tenofovir-diphosphate.

PrEP use at the second interview was highly correlated with PrEP use at Interview 1 (S1 Fig). TFV-DP levels greater than 199 fmol/punch (the lower limit of quantification) at Interview 2 were seen almost exclusively among AGYW with levels >199 fmol/punch at Interview 1. Very few AGYW who were not taking PrEP at baseline (TFV-DP <10 fmol/punch) achieved quantifiable levels at Interview 2, and almost all the AGYW who were lost-to-follow up were not taking PrEP at baseline.

Among AGYW who said they discontinued PrEP at Interview 2, primary reasons for discontinuation included stock outs of PrEP, no longer being at risk for HIV (mainly attributed to currently trusting their partner), and returning to school or being away from home (S2 Table). Side effects were also cited fairly frequently as a reason for discontinuation, but reports of discontinuation due to health effects, lifestyle demands, difficulty taking daily pills, pill characteristics, or pressure from partners were not common. Among those who had a late or missed refill visit ($N = 49$), the most common reasons offered were traveling away from home, forgot the appointment, or no provider was available. Almost all AGYW (179/197, 90.9%) who reported continuing PrEP at Interview 2 said their strong commitment to preventing HIV infection made it easy to take PrEP, and the majority (106/197, 53.8%) said that they had had no difficulty taking PrEP (S3 Table). Only a few (25/197, 13%, [95% CI 8 to 17]) of these AGYW reported not taking any PrEP in the past week. Most (166/197, 84.3%, [95% CI 79.1 to 89.3]) reported taking sufficient (4+) pills to provide protection.

Interestingly, although non-protective TFV-DP levels were detected in almost all participants, most (179/197, 91%) AGYW who said they were continuing PrEP indicated that they were taking PrEP to avoid HIV infection and believed themselves to be protected. At the second interview, participants were asked, "What do you think your chance is of getting HIV?"; only 11/197 (5%) who said they were continuing PrEP answered moderate-to-high chance, in contrast to 38/105 (36%) among discontinuers (S3 Table). AGYW who reported continuing PrEP were then asked, "What would your chance of getting HIV be if you were not taking PrEP?"; 167/197 (85%) answered moderate-to-high.

Despite attending the follow-up and PrEP refill visits and stating that they were continuing PrEP, most PrEP persisters were not adherent. Only 28/176 (16%) had detectable levels at Interview 2, with only 7 (4%) having levels consistent with protection ($\geq$700 fmol/punch). While interventions to improve adherence may benefit these AGYW, it is unclear that such interventions will help the larger group of AGYW who persist in the program without taking PrEP at all. Compared to AGYW who discontinued PrEP, this persistent but non-adherent group ($N$ = 144) was older (adjusted OR 2.13 [95% CI 1.01 to 4.49], $p$ = 0.048), more likely to be active in the DREAMS program (adjusted OR 3.85, [95% CI 1.07 to 13.82], $p$ = 0.039), think they would be at moderate-to-high risk for HIV if not taking PrEP (adjusted OR 10.17, [95% CI 5.14 to 20.13], $p < 0.001$), and more likely to use injectable contraceptives (adjusted OR 3.28, [95% CI 1.4 to 7.67], $p$ = 0.006) (Table 2).

Among AGYW who persisted in the program, no significant differences were seen between those with TFV-DP levels above versus below 10 fmol/punch in general (S4 Table), likely due to the small number ($N$ = 32) with levels above 10 fmol/punch. Models comparing all persistent AGYW ($N$ = 176) to (1) AGYW who discontinued PrEP ($N$ = 105); and (2) AGYW who attended the second interview but were non-persistent ($N$ = 105 + 21) gave similar results, although the association with partners who were supportive was no longer significant (S5 and S6 Tables).

## Discussion

In this study designed to examine PrEP adherence, persistence, and factors associated with persistence, nested within a real-world PrEP program in western Kenya, AGYW had moderate retention in the program, and most reported high adherence and continuation of oral PrEP. However, a minority had detectable TFV-DP at interview visits, and only a small percentage achieved or sustained sufficient drug concentrations to prevent HIV acquisition. Most (85%) continued to receive HIV prevention support through the DREAMS initiative. AGYW who stopped taking PrEP were more likely to have stopped participation in DREAMS (27% versus 8%) and gave a variety of reasons for discontinuation, most of which were not directly related to the PrEP product itself (stockouts, no longer being at risk, returning to school, travel away from the site, etc.). Notably, AGYW who persisted in the PrEP program had a higher self-perceived risk of HIV infection. Among AGYW who reported they were continuing PrEP at the second interview, over 90% indicated the reason was to prevent HIV, although almost all had non-protective TFV-DP levels. This study documents the disconnect between intention and action in a real-world PrEP program and identifies AGYW who may be optimal candidates for long-acting injectable PrEP.

Other studies have observed vast discrepancies between self-reported and objective PrEP adherence measures in clinical trial settings [8,10,33,34]. Importantly, adherence in our study participants was much lower than adherence in AGYW enrolled in clinical trials or in a recent trial of support strategies for oral PrEP in South Africa and Zimbabwe (89% retention at 3 months; 84% of participants tested had detectable TFV-DP) [35,36]. This suggests that oral PrEP adherence among AGYWs in real-world programmatic settings in SSA may be lower than predicted by trial settings and may not assure the drug levels necessary to prevent HIV acquisition. A recent evaluation of the DREAMS Initiative in Kenya and South Africa reported a decline in HIV incidence in AGYW which began prior to DREAMS and which did not accelerate during the first 3 years of the program [37]. Although the reasons for the failure of DREAMS to impact HIV incidence in these locales are unknown, low uptake of interventions by participants may have contributed.

The results of this observational study also provide additional data on the experiences of AGYW using PrEP in a programmatic setting in SSA. Despite their commitment to preventing

**Table 2. Factors associated with Persistence despite poor adherence (Discontinuers vs. Persisters who did not take PrEP).**

| Factor | Discontinuers (N = 105) n | Persisters taking no PrEP * (N = 144) n | Univariable analysis[1] Odds ratio OR [95%CI] | p-Value | Multivariable analysis[2] Odds ratio OR [95% CI] | p-Value |
|---|---|---|---|---|---|---|
| **Age ≥22 years** | 56 | 100 | 1.94 [1.1, 3.44] | 0.022 | 2.13 [1.01, 4.49] | 0.048 |
| Still active in the DREAMS program | 77 | 132 | 4.08 [1.74, 9.59] | 0.001 | 3.85 [1.07, 13.82] | 0.039 |
| Currently has a sexual partner | 100 | 140 | 1.72 [0.41, 7.23] | 0.456 | | . |
| Currently has multiple sex partners | 7 | 23 | 2.93 [1.09, 7.86] | 0.033 | 2.66 [0.54, 12.97] | 0.227 |
| Married/cohabiting | 44 | 67 | 1.18 [0.71, 1.97] | 0.527 | | . |
| One or more children | 68 | 110 | 1.75 [1.06, 2.88] | 0.029 | 0.96 [0.37, 2.51] | 0.931 |
| Lives with parents or grandparents | 54 | 68 | 0.86 [0.48, 1.54] | 0.610 | | . |
| Lives with partner | 43 | 66 | 1.2 [0.7, 2.04] | 0.507 | | . |
| Partner is aware of PrEP use | 51 | 88 | 1.82 [1.08, 3.06] | 0.024 | 1.95 [0.91, 4.21] | 0.088 |
| **Partner is very supportive of PrEP use** | 23 | 58 | 2.52 [1.58, 4] | <0.001 | 2.21 [1.0, 4.88] | 0.051 |
| Partner is HIV positive | 1 | 6 | 4.45 [0.72, 27.37] | 0.107 | | . |
| Partner has other partners | 16 | 41 | 2.22 [0.95, 5.16] | 0.064 | 1.13 [0.45, 2.82] | 0.796 |
| AGYW believes partner puts her at risk | 30 | 67 | 2.2 [1.16, 4.18] | 0.016 | 0.84 [0.35, 2.04] | 0.698 |
| **Moderate-to-high HIV chance if not taking PrEP** | 38 | 123 | 10.25 [5.31, 19.8] | <0.001 | 10.17 [5.14, 20.13] | <0.001 |
| Experience of intimate partner violence (IPV score >10) | 7 | 10 | 1.01 [0.39, 2.59] | 0.984 | | . |
| Depression, moderate to severe | 9 | 10 | 0.82 [0.37, 1.83] | 0.625 | | . |
| Social support (most or all the time) | 24 | 27 | 0.79 [0.45, 1.37] | 0.397 | | . |
| Inconsistent or no condom use | 82 | 116 | 1.15 [0.74, 1.78] | 0.533 | | . |
| Contraceptive use, any | 77 | 109 | 1.14 [0.64, 2.01] | 0.660 | | . |
| oral | 5 | 7 | 1.03 [0.28, 3.7] | 0.969 | | . |
| **injectable** | 13 | 30 | 1.83 [1.01, 3.33] | 0.048 | 3.28 [1.4, 7.67] | 0.006 |
| Implant | 21 | 44 | 1.76 [0.93, 3.34] | 0.083 | 1.55 [0.71, 3.39] | 0.274 |
| Male condoms | 35 | 26 | 0.44 [0.25, 0.78] | 0.004 | 1.04 [0.44, 2.45] | 0.935 |
| Female condoms | 2 | 2 | 0.7 [0.11, 4.4] | 0.705 | | . |
| Friends are on PrEP | 94 | 138 | 2.78 [0.73, 10.56] | 0.134 | | . |
| Told someone of PrEP use since Interview 1 | 45 | 76 | 1.53 [1.06, 2.2] | 0.022 | 0.9 [0.55, 1.47] | 0.684 |
| Months since PrEP initiation at Interview 1, 2–3 months | 30 | 37 | 0.88 [0.44, 1.77] | 0.725 | | . |
| Months since PrEP initiation at Interview 1, 4–6 months | 59 | 76 | 0.85 [0.57, 1.28] | 0.447 | | . |
| Months since PrEP initiation at Interview 1, 6+ months | 15 | 31 | 1.64 [0.81, 3.32] | 0.167 | | . |
| Education, primary school | 32 | 46 | 1.07 [0.61, 1.88] | 0.822 | | . |
| Education, secondary school | 57 | 85 | 1.22 [0.68, 2.2] | 0.504 | | . |
| Education, postsecondary | 16 | 13 | 0.54 [0.23, 1.31] | 0.174 | | . |
| Currently in school | 45 | 44 | 0.57 [0.38, 0.85] | 0.006 | 0.86 [0.47, 1.58] | 0.624 |
| In PrEP support group | 81 | 118 | 1.38 [0.75, 2.54] | 0.303 | | . |

* TFV-DP <10 fmol/punch.

[1]Odds ratio and corresponding p-value were based on univariable generalized estimating equations with logit link function in the model accounting for clustering of study participants within wards.

[2]Odds ratio and corresponding p-value were based on multivariable generalized estimating equations with logit link function in the model adjusted for county of residence, factors with p-value <0.1 in the univariable analysis as well as clustering of study participants within wards.

AGYW, adolescent girls and young women; PrEP, pre-exposure prophylaxis; TFV-DP, tenofovir-diphosphate.

HIV infection, few AGYW achieved high-level adherence. AGYW who discontinued PrEP use reported multiple reasons for doing so. Although stockouts were most commonly cited, this reason was cited by only a quarter of AGYW who discontinued. Many of the other reasons cited pertained to the PrEP program but had little to do with the PrEP product itself. Few AGYW cited fear of or experience with side effects (15.2% of AGYW who discontinued and 7.6% of those who said they were continuing PrEP). Some AGYW may have engaged in the HIV prevention program, while not adhering to a daily pill regimen, due to social desirability. However, their responses to questions suggest that they were motivated at least in part by perceived HIV risk. The relationship between perceived HIV risk and actual risk in discontinuers versus continuers is complex, as both groups had similar proportions who reported having a sexual partner, living with a partner, using condoms, etc., although continuers were more likely to have multiple partners (7% versus 15%) or to believe that their partner's behavior put them at risk (29% versus 44%). Only 36% of AGYW who discontinued PrEP reported their current HIV risk as moderate-to-high, although they were not taking PrEP. Only 6% of AGYW who said they were continuing PrEP assessed their current risk as moderate-to-high; a striking 79% said their current risk was low but would be moderate-to-high if they were not taking PrEP. Thus, most AGYW who said they were continuing PrEP acknowledged moderate-to-high HIV risk and the vast majority said they were using PrEP to prevent HIV. The fact that almost none had protective levels implies that these AGYW did not understand or acknowledge that only high-level adherence to daily oral PrEP affords protection.

Overall, these results reveal a group of AGYW who need tailored HIV prevention, including methods such as long-acting injectable PrEP. In our cohort, use of injectable contraception or implants was common. At the second interview, nearly half (45%) of all AGYW and 88% of those using a highly effective contraception method reported use of injectables or implants, which suggests that injections or implants may be acceptable for HIV prevention as well. While attention has focused on improving adherence to oral PrEP, understanding women who are persistent but not adherent and providing them with long-acting formulations may be a more effective way to reduce population-level HIV incidence. In addition, newer prevention modalities will need appropriate behavioral counseling support to assure understanding and correct use.

Limitations of this study include both size and generalizability, as we included a relatively small sample size from a large programmatic project, which may be most generalizable to AGYW in western Kenya. In addition, all AGYW in the study were recruited from the DREAMS initiative and were receiving support for HIV prevention that would likely not be available to them otherwise. Other limitations include potential social desirability bias, recall error in the self-reported data, and possibly remaining confounding bias. A strength was the testing of TFV-DP by LC-MS/MS as an objective measure of adherence. All AGYW participated in face-to-face interviews with a trained counselor. In addition to the self-reported data on factors that might influence PrEP use, we also had access to programmatic data, for example, on pharmacy refills.

In conclusion, our study provides insight into oral PrEP use in a real-world programmatic setting, as well as predictors of PrEP persistence in this context. The results reveal that most AGYW may be better protected by long-acting injectable PrEP than by oral daily PrEP. Future research is needed to clarify whether persistence without adequate adherence is as common among AGYW in other settings, and whether new long-acting PrEP formulations, adequately supported by facilitators identified during use of existing PrEP agents, can afford higher level protection to this very vulnerable population.

## Supporting information

**S1 STROBE Checklist. STROBE checklist.**
(PDF)

**S1 Protocol. Prospective protocol of the study.**
(DOCX)

**S1 Table. Number of participants by TFV-DP levels at Interviews 1 and 2.**
(DOCX)

**S2 Table. Reason for stopping PrEP among AGYW reporting PrEP discontinuation and experiences taking PrEP among AGYW reporting PrEP continuation at the second interview.**
(DOCX)

**S3 Table. Self-perceived risk for HIV infection among PrEP continuers versus discontinuers.**
(DOCX)

**S4 Table. Characteristics of Persisters taking PrEP versus Persisters taking no PrEP at the second interview.**
(DOCX)

**S5 Table. Factors associated with Persistence (Discontinuers versus Persisters).**
(DOCX)

**S6 Table. Factors associated with Persistence (Non-persisters versus Persisters).**
(DOCX)

**S1 Fig. TFV-DP levels among study participants at the first interview (Time point 1) and disposition at the time of the second interview (Time point 2).**
(DOCX)

## Acknowledgments

The authors would like to thank Sahar Zangeneh, Eline Appelmans, Siavash Pasalar, Lili Peng, and Jinru Tao for their early inputs into this research plan and design. We also thank all AGYW who participated in the study and all the staff involved in the data collection for the project.

## Author Contributions

**Conceptualization:** Ying Qing Chen, Ann Duerr.

**Data curation:** Jean de Dieu Tapsoba, Christopher Obong'o, Martha Brady, Edward Kariithi, Rael Obanda, Daniel Oluoch-Madiang, Paul Drain.

**Formal analysis:** Jean de Dieu Tapsoba, Ann Duerr.

**Funding acquisition:** Ying Qing Chen, Ann Duerr.

**Investigation:** Ann Duerr.

**Methodology:** Jean de Dieu Tapsoba, Jane Cover, Christopher Obong'o, Paul Drain, Ann Duerr.

**Project administration:** Kira Mori, Rael Obanda, Daniel Oluoch-Madiang, Ann Duerr.

**Resources:** Tim R. Cressey, Gordon Okomo, Edward Kariithi, Rael Obanda, Daniel Oluoch-Madiang, Paul Drain.

**Supervision:** Ying Qing Chen, Ann Duerr.

**Validation:** Jean de Dieu Tapsoba, Jane Cover, Christopher Obong'o, Ying Qing Chen, Ann Duerr.

**Visualization:** Jean de Dieu Tapsoba.

**Writing – original draft:** Jean de Dieu Tapsoba, Jane Cover, Christopher Obong'o, Martha Brady, Paul Drain, Ann Duerr.

**Writing – review & editing:** Jean de Dieu Tapsoba, Jane Cover, Christopher Obong'o, Martha Brady, Ying Qing Chen, Paul Drain, Ann Duerr.

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
