## [Editor Report · Decision Letter 0]

15 Mar 2022

Dear Dr Duerr, 

Thank you for submitting your manuscript entitled "Persistence in a PrEP program without protective drug levels is common among adolescent girls and young women in Kenya – would long-acting PrEP formulations serve this population better?" for consideration by PLOS Medicine.

Your manuscript has now been evaluated by the PLOS Medicine editorial staff as well as by an academic editor with relevant expertise and I am writing to let you know that we would like to send your submission out for external peer review.

Before we proceed, please be aware that the in-house editorial staff noted that you conducted research in a foreign country. Please check the relevant national regulations and laws applying to foreign researchers and state whether you obtained the required permits and approvals. PLOS seeks to uphold equity and inclusivity in global health research including authorship. In light of this, please clarify why a local researcher is not a first, second or senior author on this paper, and state measures taken to ensure inclusive participation in the writing of this manuscript. Please ensure authorship criteria is based on the International Committee of Medical Journal Editors (ICMJE) Uniform Requirements for Manuscripts Submitted to Biomedical Journals.

Please re-submit your manuscript within two working days, i.e. by Mar 17 2022 11:59PM.

Kind regards,

Beryne Odeny

PLOS Medicine

---

## [Decision Letter · Decision Letter 1]

20 Apr 2022

Dear Dr. Duerr,

Thank you very much for submitting your manuscript "Persistence in a PrEP program without protective drug levels is common among adolescent girls and young women in Kenya – would long-acting PrEP formulations serve this population better?" (PMEDICINE-D-22-00813R1) for consideration at PLOS Medicine. 

Your paper was evaluated by a senior editor and discussed among all the editors here. It was also discussed with an academic editor with relevant expertise, and sent to independent reviewers, including a statistical reviewer.

Considering these reviews, we would be grateful if you could please revise your manuscript to respond to comments raised by reviewers. We would strongly recommend that you pay special attention to the statistical reviewers’ comments. Please note that this is not a guarantee that we will accept the manuscript and that further consideration is dependent on the submission of a manuscript that addresses all reviewer concerns. We will carefully review your manuscript upon revision, so please ensure that your revision is thorough.

The reviews are appended at the bottom of this email and any accompanying reviewer attachments can be seen via the link below:

[LINK]

In light of these reviews, I am afraid that we will not be able to accept the manuscript for publication in the journal in its current form, but we would like to consider a revised version that addresses the reviewers' and editors' comments. Obviously we cannot make any decision about publication until we have seen the revised manuscript and your response, and we plan to seek re-review by one or more of the reviewers. 

We expect to receive your revised manuscript by May 11 2022 11:59PM. Please email us (plosmedicine@plos.org) if you have any questions or concerns.

We look forward to receiving your revised manuscript. 

Sincerely,

Beryne Odeny, 

PLOS Medicine

plosmedicine.org

1) Please revise your title according to PLOS Medicine's style. Your title must be nondeclarative and not a question. It should begin with main concept if possible. Please place the study design in the subtitle (i.e., after a colon), e.g., a prospective cohort study.

2) Please include line numbers in your next draft.

3) Under the list of authors please include a footnote that indicates that the first 3 authors are joint first authors, in keeping with your prior request to make them co-/first authors.

4) The Data Availability Statement (DAS) requires revision. For each data source used in your study: 

a) If the data are owned by a third party but freely available upon request, please note this and state the owner of the data set and contact information for data requests (web or email address). Note that a study author cannot be the contact person for the data.

b) If the data are not freely available, please describe briefly the ethical, legal, or contractual restriction that prevents you from sharing it. Please also include an appropriate contact (web or email address) for inquiries (again, this cannot be a study author).

5) Abstract:

a) Please structure your abstract using the PLOS Medicine headings (Background, Methods and Findings, Conclusions).

i) Please combine the Methods and Findings sections into one section, “Methods and findings”.

ii) The interpretation should be renamed “Conclusions”

b) Please include years during which the study took place

c) Please include the important dependent variables that are adjusted for in the analyses.

d) Please ensure that all numbers presented in the abstract are present and identical to numbers presented in the main manuscript text.

e) Please quantify the main results (please present both 95% CIs and p values).

f) In the last sentence of the Abstract Methods and Findings section, please describe the main limitation(s) of the study's methodology.

g) Please remove “funding” statement

7) Introduction: 

a) Please remove the description of the methods from the last paragraph, i.e., “we examine PrEP use cascade… refill data from the DREAMS database.” This can be moved to the Methods section.

b) Please conclude the Introduction with a clear description of the study question or hypothesis.

8) Did your study have a prospective protocol or analysis plan? Please state this (either way) early in the Methods section. 

9) Please ensure that the study is reported according to the STROBE and include the completed STROBE checklist as Supporting Information. Please add the following statement, or similar, to the Methods: "This study is reported as per the Strengthening the Reporting of Observational Studies in Epidemiology (STROBE) guideline (S1 Checklist)."

10) Statistical analysis

a) Please include adjustment variables. 

b) Please adjust for clustering at individual and facility level.

11) Please provide p values in addition to 95% CIs in the main text and tables

12) Please define all abbreviations in Tables and Figures e.g., PrEP, LTFU, TFV-DP, DBS, AGYW

13) Please indicate in the figure caption the meaning of the bars and whiskers in Figure S1.

14) Please replace the term “condom users” with “AGYW who use condoms”

15) Please remove the ‘Declaration of interest statement” and “Data sharing” from the end of the main text. In the event of publication, this information will be published as metadata based on your responses to the submission form.

16) References:

a) Please select the PLOS Medicine reference style in your citation manager. In-text reference call outs should be presented as follows noting the absence of spaces within the square brackets, e.g., "... services [1,2]."

b) References should have six names before et al. For those with more than six names, please ensure that et al., is inserted after six names

c) Please ensure that journal name abbreviations consistently match those found in the National Center for Biotechnology Information (NCBI) databases. https://journals.plos.org/plosmedicine/s/submission-guidelines#loc-references. 

d) Ref #20 is incomplete

Comments from the reviewers:

Reviewer #1: The manuscript by Tabsoba, et al. report on the "real-world" use of oral PrEP among AGYW in two counties in western Kenya with a high incidence of HIV. Please see my detailed comments below about each section.

Introduction: 

* The statement on new HIV infections in the two countries comes from a 2015 reference, seven years ago. The authors should update this number with a more recent reference.

* I would not consider the 3-month visit "long-term adherence"

* How were the 336 participants selected from the larger group in the DREAMS PrEP program—either explain in introduction, or remove this detail and leave for the methods section

Methods:

* This sentence doesn't make sense to me: "and a two-level factor potentially associated with PrEP adherence."

* I realize that this study was conducted in a real-world setting, but anticipating a 50% loss to follow-up is very high

* I'm confused about the methods of sampling. In one section it is mentioned that a random sample was obtained from the original enrolled cohort in the DREAMS PrEP program. However, the methods used to randomly sample are not provided. Secondly, a flow chart would help the reader to assess what happened along the enrolment cascade, who was subsequently excluded from the analysis (e.g. they failed to return for follow-up, and/or didn't report taking PrEP) would be tremendously helpful. In the Results section (first paragraph) there is new mention about who was eligible—higher risk sexual behaviors. This information makes once think that random sampling was not deployed. I think that these are the criteria that were used to enroll the DREAMS cohort, but can't be certain. Again, this section should be in the Methods not the Results section.

Results:

* "Kisumu, the remainder in Homa Bay." Does this refer to the city/town or counties?

* "PrEP program for a mean of 142 days." Provide SD—or is median a better measure with IQR?

* Throughout Results it would be helpful to the reader to provide the Ns along with the percentages.

* "PrEP persistence," not sure this is the standard designation in the literature

* Suggest adding "among" to "in contrast to 38/105 (36%) 'among' discontinuers."

* Table 1 has quite several errors—second column is not all %s, use of "****", etc. It also could benefit from reformatting

* Why was the US Household Security Survey used? The validated HFIAS has been used repeatedly in the same population in western Kenya

* I'm not sure that Figure 2 adds much more than what is stated in the narrative

Discussion:

* The discussion section is well written in general, with adequate references to substantiate the relevance of the new data from the study. In addition, the authors do spell out some of the weaknesses, while acknowledging its strengths. My primary concern is with generalizability of these data, based on some of the weaknesses I identified with the methods, or at least the description of the methods used.

Reviewer #2: Statistical review

This paper reports an observational study that investigates factors associated with persistence in a PrEP programme amongst adolescent girls and young women. I had some comments on the statistical methods/reporting of the study:

1. Abstract "samples taken at the second visit" - it might be useful if the same language as the previous sentence is used - is second visit the same as the second interview or the interim refill visit?

2. Methods, analysis "Persistent AGYW were defined as having attended both interviews as well as interim visits for PrEP refills and stating that they were taking PrEP at both interviews. Those lost to follow up between visits were excluded from analysis." - I found this confusing whether those lost to follow-up were included as non-persistent or excluded, or is this saying that only persistent AGYW were included in analysis? If loss to follow-up led to exclusion, this seems to mean an important set of likely non-persisters was excluded?

3. Methods, analysis: can more be said about what characteristics were considered for the model - is it the ones in study procedures? Some of these wouldn't vary between visit 1 and visit 2 but others would - I wasn't sure why visit 2 characteristics were used instead of visit 1 characteristics? Using visit 1 characteristics would strike me as being more about predicting which participants are persistent. 

4. Methods, analysis: can more be added about the stepwise procedures used to build the model (including rules for adding/deleting variables), perhaps in supplementary material? Standard approaches such as forward or backward selection have been criticised so it would be good if more than 1 approach led to the same variables being included, to demonstrate robustness. It may be worth considering a sparse regression approach to supplement this analysis.

5. Methods, analysis: in the results, censored individuals are mentioned - presumably these were excluded from analysis?

6. Table 2 and Figure 2 - I didn't follow why the p-values were calculated comparing discontinuers and persisters who were taking no PrEP. This doesn't seem consistent with the hypothesis that the study was powered for in the sample size calculation: factors related to PrEP adherence.

7. I think Figure S1 would be better included in the main paper.

James Wason

Reviewer #3: Thank you for the opportunity to review this manuscript that describes an assessment of PrEP adherence in 336 AGYW participating in a DREAMS programme in Western Kenya who attended two study visits. 

Major comments

1. The manuscript would benefit from a figure that shows the study design and or the attrition of participants at each step of the analysis. Participants appear to be excluded at certain points but the rationale for non-inclusion was not entirely clear as those with some information may provide important insights.

2. The primary outcome appears to be PrEP adherence defined as TDF/FTC levels on DBS >700f/ml per punch. The authors appear to use a variety of cut-offs however and it is not always clear how these were chosen. In the methods it would be helpful to provide a brief summary of the cut offs and the reasons selected based on published evidence as well as how these correspond with dosing (and limitations given largely based on MSM data). DBS reflects use in the past 4 weeks, but cut-offs may vary depending on whether this is first 4 weeks of use or not. The authors may want to be more clear about these time frames and the relation between outcome assessments and timeframes for reporting exposures/factors associated with adherence.

3. The authors argue that they have defined a PrEP cascade; however the definitions are somewhat different to definitions published by other authors on the PrEP cascade. It would be helpful if the authors could better define the process of the DREAMS PrEP programme, e.g. how initiation occurs, how often refills are given, frequency of HIV testing etc. Then the authors can locate their study on top of this framework and provide their definition of persistence which appears to be related to study visits primarily, with the requirement to return for at least one PrEP refill. The authors could perhaps explain why visits were not aligned with PrEP refill visits for AGYW as that could assist with anchoring their recall/reporting better perhaps. The authors may wish to include a brief review of PrEP cascades in the introduction to underpin their choice of terminology or methods.

4. The authors use several measures in the baseline assessment as well as for assessments at the second visit. These are not defined in the methods but only as footnotes to tables. It would be helpful to define these measures more clearly in the methods section and reasons for use/validation of these measures in adolescent populations in sub-Saharan Africa. For example, is it appropriate to use a US-based food security measure in the Kenyan context? When defining measures add scales, cut-offs and whether or not validated in this or similar young African populations.

5. The authors do not describe the package of interventions offered to AGYW to support their PrEP adherence apart from a brief mention in table 1 regarding a PrEP support group. There is value in understanding what if any adherence support interventions were offered and how they aligned with PrEP refill visits to provide context to the environment in which PrEP was offered.

6. The authors used an interviewer administered tool to solicit information from participants. Could they comment further of strategies to minimise social desirability bias. The participants seem to understand that they were being interviewed about their PrEP use. The authors observe the difference between reported and actual actions, but do not comment on the potential bias induced by the data collection process.

7. In the discussion, the authors appear to discount reasons for non-PrEP use as not related to PrEP. However in reviewing table S2 it appears that many of the reasons for discontinuation are supply side or programmatic issues e.g. stock outs, lack of access because travelled from PrEP refill site or returned to school. In addition, side effects may also be an issue with poor counselling for those who are in early PrEP start up. The aggregation of PrEP users at different stages of PrEP use (less/more experienced) makes it challenging to tease apart some of these issues.

8. There are very few observations in the manuscript around the social and structural barriers to PrEP access for these young participants e.g. 17% experience IPV, 12% have adequate social support, 75% have food insecurity, most are married and have children and live in multi-generational homes. It is true that the data confirms that participants are motivated to use PrEP but are not able to act on those intentions perhaps because of the more pressing concerns in their life. It would be helpful if the authors could reflect on how these upstream factors may influence the direct factors observed to be associated with lack of persistence.

9. At times the sections bled into one another, methods in the results section, results in the discussion and discussion in results. The manuscript would benefit from a thorough edit to define each section more clearly and to make it easier for the reader to follow the arguments being made with the data.

10. In table 2, it is not clear what the P-value refers to (although the heading defines persisters vs continuers). Please clarify which comparison the p-value refers to with respect to the persister column, since those with detectable vs non-detectable PrEP appear quite different. It may be more helpful to present an analysis of continuers vs. discontinuers to understand barriers to PrEP continuation, and then to explore reasons for non-adherence which may not be the same as those for continuation and may require other interventions to address. Long-acting agents may help with the adherence issue, but may still be perplexed with the discontinuation challenge.

11. T

Minor comments

12. Abstract: define the outcome measures and cut offs for DBS as well as the assay in the methods, and the definitions used for levels consistent with no dosing/levels associated with protection.

13. Abstract: define the persistence outcome in the methods of the abstract

14. Abstract: the conclusion states that this analysis identifies AGYW who may benefit from LA PrEP but it is not entirely clear from the findings how this population are identified without the use of a drug level assessment. Please clarify

15. Introduction: TDF/FTC = tenofovir disoproxil fumarate/emtricitabine. Please spell out in full

16. Introduction: the authors should mention the PK requirement for near perfect PrEP dosing in women which may compound challenges with adherence and are different for MSM at least.

17. Introduction: please provide references for obstacles to adherence support in para 1 e.g. lack of partner support.

18. Introduction: CAB LA was evaluated in cisgender men and transgender women who has sex with men in the Americas, Thailand and South Africa and both HPTN 083 and 084 showed that CAB LA was superior to TDF/FTC for HIV prevention.

19. Introduction: clarify that plasma TDF/FTC was detectable in a random subset 56% of samples in HPTN 084 in women who had a median follow up of 1.2 years. The time frame seems relevant. In addition, injection coverage was across both arms of the study not just the CAB LA group. The sentence about visit attendance vs adherence probably needs to conclude this paragraph 

20. Introduction: define persistence in relation to literature

21. Introduction: consider introducing a summary of the notion of a PrEP cascade and mention of the concept of Prevention effective adherence

22. Introduction: final comment appears to bleed into methods and may be better as a framing of objectives with definitions in methods

23. Methods: would benefit from a figure that summarises time on PrEP and relation to sample collection and two visits/could summarise the reasons for inclusion/exclusion from analysis at each step of design

24. Methods: when defining measures add scales, cut-offs and whether or not validated in this or similar young African populations

25. Methods: what was the interval between study visits?

26. Results: para 1; a description of the DREAMS PrEP programme including eligibility criteria for PrEP may be better in the methods since the reasons for non-inclusion are not related to these criteria it seems.

27. Results: can you provide some distribution of where respondents were in their PrEP journey with respect to the mean 142 days of PrEP? What % were in first month, months 2-3, months 3-6 as it might relate to a PrEP programme. This makes it easier then also to overlay where discontinuation might be happening.

28. Results: what was the mean interval between study visits?

29. Results: para 2 provide numerator, denominator and percentages so that it is possible to follow the logic of which group is being discussed.

30. Results: clarify lower limit of quantification stated as >199 f/mol per punch - this seems to contradict other statements and measures and may not be correct. Should this be defined in the methods?

31. Results: para that begins " Thus it seems…" seems to be editorial commentary and may be better as a comment in the discussion with a focus on presenting the outcomes in the results.

32. Discussion: its not clear what data support the statement that programme retention was good.

33. Discussion: the clinical trial referred to in para 2 was not a trial of PrEP but an assessment of uptake and acceptability and included an evaluation of adherence strategies. Perhaps the difference is not real world vs. not real word but the focus on adherence support for an adolescent population. 

34. Discussion: para 3, not clear what data support the statement that this shows a cascade of care in the sense of how cascades have been described in the literature. May need to refine or clarify.

35. Discussion: the authors mention that they had access to pharmacy refill data. It would be helpful to summarise these for the cohort and present in relation to when they were interviewed. The refill data are mentioned but data are not presented systematically.

36. References: update to reflect published manuscripts where possible; some conference abstracts are referred to when in fact peer-reviewed manuscripts are available.

[LINK]

---

## [Decision Letter · Decision Letter 2]

21 Jul 2022

Dear Dr. Duerr,

Thank you very much for re-submitting your manuscript "Continued attendance in a PrEP program despite low adherence and non-protective drug levels among adolescent girls and young women in Kenya: results from a prospective cohort study" (PMEDICINE-D-22-00813R2) for review by PLOS Medicine.

I have discussed the paper with my colleagues and the academic editor and it was also seen again by one reviewer. I am pleased to say that provided the remaining editorial and production issues are dealt with we are planning to accept the paper for publication in the journal.

[LINK]

We look forward to receiving the revised manuscript by Jul 28 2022 11:59PM.   

Sincerely,

Beryne Odeny, 

PLOS Medicine

plosmedicine.org

Requests from Editors:

1. Abstract: please include the important dependent variables that are adjusted for in the analyses.

2. Author summary: please quantify the main results with actual amounts or percentages

3. Prospective protocol:

a) In the Methods, please indicate whether this study had a prospective protocol and identify changes that were made in this analysis including those in response to peer review.

b) Please include the relevant prospectively written document with your revised manuscript as a Supporting Information file to be published alongside your study and cite it in the Methods section. A legend for this file should be included at the end of your manuscript.

Comments from Reviewers:

Reviewer #2: Thank you to the authors for addressing my previous comments well. All the issues I raised were addressed/justified and I have no further issues to raise.

[LINK]

---

## [Editor Report · Decision Letter 3]

12 Aug 2022

Dear Dr. Duerr,

Thank you very much for re-submitting your manuscript "Continued attendance in a PrEP program despite low adherence and non-protective drug levels among adolescent girls and young women in Kenya: results from a prospective cohort study" (PMEDICINE-D-22-00813R3) for review by PLOS Medicine.

I have discussed the paper with my colleagues and I am pleased to say that provided the remaining editorial and production issues are dealt with we are planning to accept the paper for publication in the journal.

[LINK]

We look forward to receiving the revised manuscript by Aug 19 2022 11:59PM.   

Sincerely,

Beryne Odeny, 

PLOS Medicine

plosmedicine.org

Requests from Editors:

1. Author summary: Please provide estimates as amounts and percentages. Please do not use coefficients and p-values in the author summary as this is supposed to be a non-technical summary for a general audience.

[LINK]

---

## [Editor Report · Decision Letter 4]

19 Aug 2022

Dear Dr Duerr, 

On behalf of my colleagues and the Academic Editor, Dr. Marie-Louise Newell, I am pleased to inform you that we have agreed to publish your manuscript "Continued attendance in a PrEP program despite low adherence and non-protective drug levels among adolescent girls and young women in Kenya: results from a prospective cohort study" (PMEDICINE-D-22-00813R4) in PLOS Medicine.

PRESS

Sincerely, 

Beryne Odeny, PhD 

PLOS Medicine